# Long-term therapeutic silencing of miR-33 increases circulating triglyceride levels and hepatic lipid accumulation in mice

Leigh Goedeke[1,2,3,4,†], Alessandro Salerno[3,4,†], Cristina M Ramírez[1,2,3,4], Liang Guo[3,4], Ryan M Allen[5], Xiaoke Yin[6], Sarah R Langley[6], Christine Esau[7], Amarylis Wanschel[3,4], Edward A Fisher[3,4], Yajaira Suárez[1,2,3,4], Angel Baldán[5], Manuel Mayr[6] & Carlos Fernández-Hernando[1,2,3,4,*]

## Abstract

Plasma high-density lipoprotein (HDL) levels show a strong inverse correlation with atherosclerotic vascular disease. Previous studies have demonstrated that antagonism of miR-33 *in vivo* increases circulating HDL and reverse cholesterol transport (RCT), thereby reducing the progression and enhancing the regression of atherosclerosis. While the efficacy of short-term anti-miR-33 treatment has been previously studied, the long-term effect of miR-33 antagonism *in vivo* remains to be elucidated. Here, we show that long-term therapeutic silencing of miR-33 increases circulating triglyceride (TG) levels and lipid accumulation in the liver. These adverse effects were only found when mice were fed a high-fat diet (HFD). Mechanistically, we demonstrate that chronic inhibition of miR-33 increases the expression of genes involved in fatty acid synthesis such as acetyl-CoA carboxylase (ACC) and fatty acid synthase (FAS) in the livers of mice treated with miR-33 antisense oligonucleotides. We also report that anti-miR-33 therapy enhances the expression of nuclear transcription Y subunit gamma (NFYC), a transcriptional regulator required for DNA binding and full transcriptional activation of SREBP-responsive genes, including ACC and FAS. Taken together, these results suggest that persistent inhibition of miR-33 when mice are fed a high-fat diet (HFD) might cause deleterious effects such as moderate hepatic steatosis and hypertriglyceridemia. These unexpected findings highlight the importance of assessing the effect of chronic inhibition of miR-33 in non-human primates before we can translate this therapy to humans.

**Keywords** cholesterol; fatty acids; hepatic steatosis; microRNA
**Subject Categories** Metabolism; Pharmacology & Drug Discovery

## Introduction

Non-alcoholic fatty liver disease (NAFLD) is the most common form of chronic liver disease worldwide and represents a clinicopathologic condition characterized by lipid deposition in hepatocytes of the liver parenchyma in the absence of heavy alcohol consumption (Samuel *et al*, 2010; Samuel & Shulman, 2012). The major risk factors for developing NAFLD are associated with insulin resistance, such as obesity, type II diabetes, dyslipidemia, and the metabolic syndrome. Insulin resistance affects carbohydrate and lipid metabolism, increasing triglyceride accumulation in the liver (Samuel *et al*, 2010; Samuel & Shulman, 2012). While this lipid buildup can be caused by several metabolic imbalances, such as decreased free fatty acid (FFA) oxidation or decreased triglyceride (TG) export, the largest contributor to hepatic steatosis in patients with NAFLD is increased FFA delivery to the liver and increased hepatic *de novo* lipogenesis (DNL).

Several families of transcription factors, including sterol regulatory element-binding proteins (SREBPs), regulate the expression of genes involved in lipid metabolism in the liver (Brown & Goldstein, 1997; Horton *et al*, 2002). The SREBP family of basic-helix-loop-helix zipper (bHLH-LZ) transcription factors consists of the SREBP1a, SREBP1c, and SREBP2 proteins that are encoded by two unique genes, *SREBP1* and *SREBP2* (Brown & Goldstein, 1997; Horton *et al*, 2002). The SREBPs differ in their tissue-specific expression, target gene selectivity, and relative potencies of their trans-activation domains. SREBP2 regulates the expression of genes involved in controlling cholesterol homeostasis, such as 3-hydroxy-3-methylglutaryl coenzyme A reductase (HMGR) and the low-density lipoprotein receptor (LDLR) (Osborne, 2000). SREBP1c is the predominant isoform in the liver and positively regulates the expression of genes encoding lipogenic enzymes, including acetyl-CoA carboxylase (ACC) and fatty acid synthase (FAS). Specifically, SREBP1-mediated lipogenesis has been

1 Vascular Biology and Therapeutics Program, Yale University School of Medicine, New Haven, CT, USA
2 Integrative Cell Signaling and Neurobiology of Metabolism Program, Section of Comparative Medicine, Yale University School of Medicine, New Haven, CT, USA
3 Leon H. Charney Division of Cardiology, Department of Medicine, New York University School of Medicine, New York, NY, USA
4 Marc and Ruti Bell Vascular Biology and Disease Program, New York University School of Medicine, New York, NY, USA
5 Edward A. Doisy Department of Biochemistry and Molecular Biology, Center for Cardiovascular Research, Saint Louis University School of Medicine, Saint Louis, MO, USA
6 King's British Heart Foundation Centre, King's College London, London, UK
7 Regulus Therapeutics, San Diego, CA, USA
*Corresponding author. Tel: +1 203 737 4615; Fax: +1 203 737 2290; E-mail: carlos.fernandez@yale.edu
†These authors contributed equally to this work

shown to correlate with fatty liver susceptibility. Insulin is a well-known stimulator of lipogenesis and activates the hepatic expression of SREBP1c (Osborne, 2000). When insulin levels are high, SREBP-1c is transcribed and processed at extremely high levels, and the resultant nuclear SREBP-1c activates genes necessary to produce fatty acids, which are incorporated into triglycerides. This ensuing elevated hepatic lipogenesis is a troublesome consequence of hyperinsulinemia observed in obesity and type II diabetes mellitus.

In addition to the classic transcriptional regulators, compelling evidence indicates that microRNAs (miRNAs) also regulate energy metabolism and liver functions (Fernandez-Hernando *et al*, 2011, 2013). miRNAs are endogenous approximately 22 nucleotide (nt) RNAs that negatively regulate mRNAs by imperfect base pairing to the 3′untranslated regions (3′UTRs) of their targets (Ambros, 2004; Bartel, 2004; Filipowicz *et al*, 2008). Recently, we and others have identified a highly conserved family of miRNAs, miR-33a/b, embedded within the intronic sequences of *SREBP* genes (Marquart *et al*, 2010; Najafi-Shoushtari *et al*, 2010; Rayner *et al*, 2010). The human genome encodes for two isoforms of miR-33: *miR-33a,* which is encoded within intron 16 of the *SREBP2* gene and *miR-33b,* which is located within intron 17 of the *SREBP1* gene. While miR-33b conservation is lost in lower mammals, including rodents, miR-33a is highly conserved from *Drosophila* to humans. Transcriptional activation of SREBP1 and SREBP2 also increases miR-33a and miR-33b levels, suggesting that miR-33a/b are regulated with their host genes (Marquart *et al*, 2010; Najafi-Shoushtari *et al*, 2010; Rayner *et al*, 2010). Both miRNAs regulate genes involved in the regulation of cellular cholesterol export, fatty acid oxidation, insulin signaling, and glucose production, including adenosine triphosphate binding cassette A1 (ABCA1), carnitine O-octanyl transferase (CROT), carnitine palmitoyltransferase 1A (CPT1A), hydroxyacyl-CoA dehydrogenase-3-ketoacyl-CoA (HADHB), AMP-activated protein kinase (AMPK), phosphoenolpyruvate carboxykinase (PCK1), and glucose-6-phosphatase (G6PC) (Rayner *et al*, 2010; Davalos *et al*, 2011; Goedeke *et al*, 2013; Ramirez *et al*, 2013). Of note, antagonism of miR-33 *in vivo* or genetic ablation of miR-33 results in a significant increase of circulating high-density lipoprotein cholesterol (HDL-C) levels (Marquart *et al*, 2010; Najafi-Shoushtari *et al*, 2010; Rayner *et al*, 2010). Similarly, non-human primates treated with anti-miR-33 oligonucleotides also exhibit increased HDL-C levels (Rayner *et al*, 2011a; Rottiers *et al*, 2013). Because HDL-C levels and increased reverse cholesterol transport (RCT) have shown a strong inverse correlation with atherosclerotic vascular disease, several groups decided to study the efficacy of anti-miR-33 therapy during the progression and regression of atherosclerosis. These studies demonstrated in most cases that antagonism of miR-33 *in vivo* delays the progression and enhances the regression of atherosclerosis (Rayner *et al*, 2011b; Marquart *et al*, 2013; Rotllan *et al*, 2013).

In addition to the established role of miR-33 in controlling plasma HDL levels, inhibition of miR-33 *in vitro* markedly increases fatty acid oxidation (Davalos *et al*, 2011), suggesting that anti-miR-33 therapy might be useful to reduce hepatic lipid accumulation and treat patients with NAFLD. Thus, in the present study, we tested the efficacy of long-term anti-miR-33 therapy on lipoprotein metabolism and the hepatic lipid profile in C57BL/6 mice fed a chow diet (CD) or high-fat diet (HFD). Surprisingly, we found that prolonged silencing of miR-33 results in hepatic lipid accumulation and increased plasma TG levels in mice fed a HFD. Importantly, hepatic expression of genes involved in fatty acid synthesis such as FAS, ACC, and SREBP1

was increased in mice treated with miR-33 antisense oligonucleotides. Proteomic analysis demonstrates that long-term anti-miR-33 therapy results in a pronounced upregulation of major urinary proteins that can make up 5% of the total RNA transcripts in the male murine liver. Finally, we found that nuclear transcription Y subunit gamma (NFYC), a miR-33 target gene, was markedly increased in mice administered with miR-33 ASO compared to control miRNA-treated mice. NFYC is a member of the three NF-Y subunits required for DNA binding and full transcriptional activation of SREBP-responsive genes (Osborne, 2000). This finding could explain why sustained derepression of miR-33 *in vivo* increases hepatic NFYC levels leading to increased expression of SREBP-regulated genes, such as FAS, ACC, HMGCR, and LDLR.

## Results

### Long-term anti-miR-33 therapy increases plasma triglyceride levels in high-fat diet fed mice

Previous short-term studies (4 weeks) showed that mice fed a chow diet (CD) and treated with anti-miR-33 oligonucleotides have increased circulating HDL-C without affecting the cholesterol distribution in other lipoproteins fractions. To determine whether long-term anti-miR-33 therapy was also efficient in increasing plasma HDL-C, we treated C57BL6 mice with 2′fluoro/methoxyethyl (2′F/MOE) phosphorothioate-backbone-modified anti-miR-33 oligonucleotides (miR-33 ASO). Similar to our previous short-term studies, miR-33 ASO-treated mice showed a marked reduction of hepatic miR-33 expression (Fig 1A) and had increased total cholesterol and HDL-C (Fig 1B and C) compared with those receiving PBS or control anti-miR (Cont ASO). The cholesterol distribution in other lipoprotein fractions, as well as triglycerides (TG) and body weight, was similar in the three groups of mice (Fig 1D–2F).

We further analyzed the effect of long-term administration of miR-33 ASO in high-fat diet (HFD) fed mice. Similar to mice fed a chow diet, hepatic miR-33 expression was significantly reduced in mice receiving miR-33 ASO compared to PBS or Cont ASO (Fig 1G). Total cholesterol and circulating HDL-C were increased in mice treated with miR-33 ASO compared with PBS and Cont ASO (Fig 1H and I). Surprisingly, plasma TG levels were also significantly elevated in mice receiving miR-33 ASO (Fig 1J). Moreover, we also found increased levels of ApoB-100, the main VLDL/LDL-associated lipoprotein, in mice treated with miR-33 inhibitors (Fig 1K). Analysis of the plasma lipoprotein distribution showed that miR-33 ASO-treated mice had a significant increase in cholesterol associated with the HDL fraction and TGs in the VLDL fraction (Fig 1L and M). The body weight was similar between the three groups of mice (Fig 1N). Collectively, these results suggest that while prolonged miR-33 ASO treatment in chow-fed mice increases circulating HDL-C without affecting plasma TG levels, long-term anti-miR-33 therapy in HFD-fed mice results in hypertriglyceridemia.

### Chronic miR-33 ASO administration results in moderate hepatic steatosis

To gain insights into the potential mechanism behind the hypertriglyceridemia observed in mice treated with miR-33 ASO, we analyzed

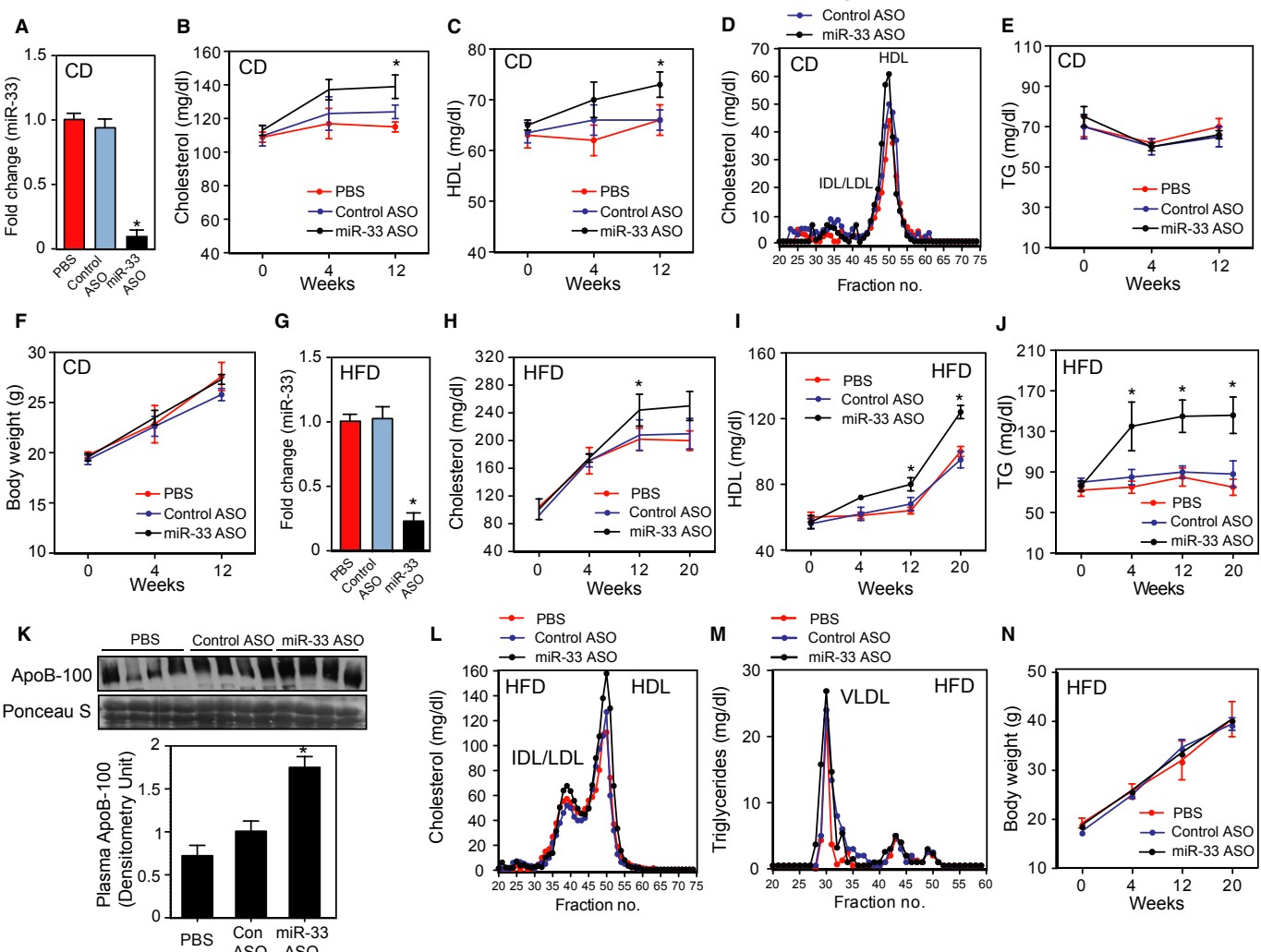

**Figure 1.  Long-term anti-miR-33 therapy results in hypertriglyceridemia in mice fed a HFD.**

A      qRT-PCR analysis of hepatic miR-33 expression levels in the livers of mice treated with PBS, control ASO, or miR-33 ASO, and fed a chow diet (CD).

B, C   Plasma cholesterol (B) and HDL-C (C) levels in the livers of mice treated with PBS, control ASO, or miR-33 ASO for 4 and 12 weeks and fed a CD.

D      Lipoprotein profile analysis obtained from pooled plasma of mice administered PBS, control ASO, or miR-33 ASO.

E, F   Circulating triglyceride (TG) levels (E) and body weight (F) of mice injected with PBS, control ASO, or miR-33 ASO, and fed a CD.

G      qRT-PCR analysis of hepatic miR-33 expression levels of mice treated with PBS, control ASO, or miR-33 ASO, and fed a high-fat diet (HFD).

H–J    Plasma cholesterol (H), HDL-C (I) and triglyceride (J) levels of mice treated with PBS, control ASO, or miR-33 ASO for 4 and 12 weeks and fed a HFD.

K      Representative Western blot of plasma ApoB-100 expression of mice treated with PBS, control ASO, or miR-33 ASO and fed a HFD for 20 weeks.

L, M   Cholesterol (L) and triglyceride (M) distribution in different lipoprotein fractions isolated from mice treated with PBS, control ASO, or miR-33 ASO, and fed a HFD.

N      Body weight of mice treated with PBS, control ASO, or miR-33 ASO for 20 weeks and fed a HFD.

Data information: All the data represent the mean ± SEM; (PBS $n = 10$, control ASO $n = 12$ and miR-33 ASO $n = 12$) and *$P < 0.05$ comparing miR-33 ASO group with PBS and control ASO groups. Lipoprotein fractionation analyses were performed using pooled plasma from five mice in each group.

Source data are available online for this figure.

the effect of anti-miR-33 therapy on hepatic lipid metabolism and gene expression. The results showed that long-term miR-33 silencing leads to a marked hepatic accumulation of TG, diglycerides (DG), free fatty acids (FFA), and cholesterol esters (CE) compared to mice injected with PBS or control ASO (Fig 2A–D). Furthermore, neutral lipid staining using Oil Red O confirmed the accumulation of lipids in mice treated with miR-33 ASO (Fig 2E). Interestingly, the hepatic lipid accumulation was only observed in miR-33 ASO-treated mice fed a HFD but not in mice fed a CD (Fig 2A–E). Liver fibrosis

assessed using pricrosirius red was similar in both groups of mice (Fig 2E).

We next analyzed lipid and glucose metabolism gene expression. As expected, anti-miR-33 therapy significantly increased the mRNA expression of previously identified miR-33 target genes, including receptor-interacting protein 140 (RIP140), CROT, CPT1A, HADHB, nuclear co-activator 3 (SRC3), AMPK, and SRC1 (Fig 2F–J). Similar results were observed at the protein level (Fig 2K). Importantly, the expression level of genes involved in fatty acid synthesis, such as

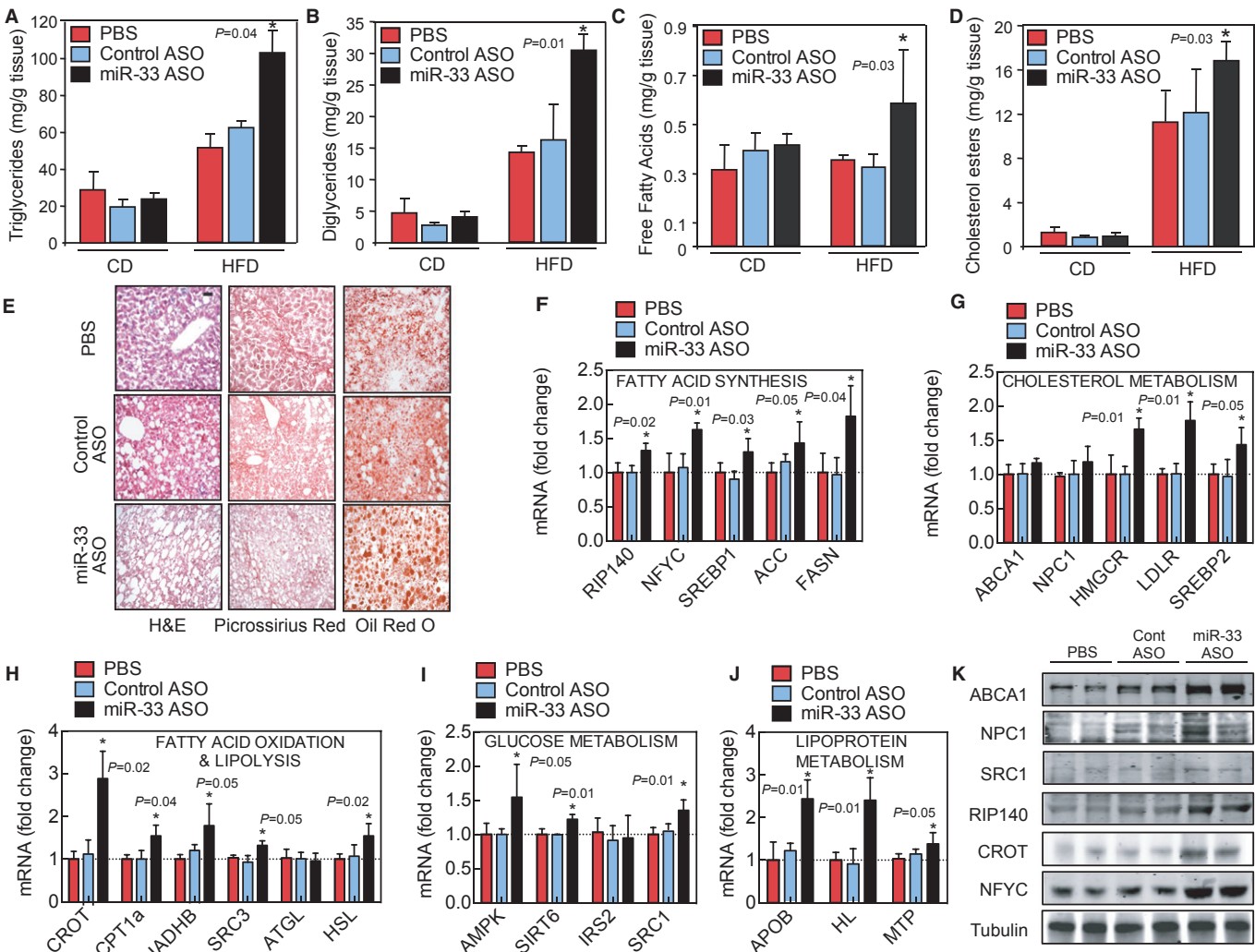

**Figure 2. Antagonism miR-33 in mice fed a HFD results in moderate hepatic steatosis.**

A–D   Hepatic content of triglycerides (A), diglycerides (B), free fatty acids (C), and cholesterol esters (D) quantified from liver tissue of mice treated with PBS, control ASO, or miR-33 ASO for 20 weeks and fed a chow diet (CD) or high-fat diet (HFD). Data represent the mean ± SEM; (PBS *n* = 3, control ASO *n* = 6 and miR-33 ASO *n* = 6) and *$P$ < 0.05 comparing PBS and miR-33 ASO group with control ASO group.

E     Representative liver sections isolated from mice treated with PBS, control ASO, or miR-33 ASO stained with H&E, picrossirius red, and Oil Red O. Scale bar = 70 μm.

F–J   qRT-PCR analysis of genes involved in fatty acid synthesis (F), cholesterol metabolism (G), fatty acid oxidation and lipolysis (H), glucose metabolism (I) and lipoprotein metabolism (J) in liver tissues from mice treated with PBS, control ASO or miR-33 ASO. The mRNA fold change from each gene from mice treated with miR-33 ASO or control ASO-treated mice compared to PBS-treated mice was calculated. Data represent the mean ± SEM; (PBS *n* = 3, control ASO *n* = 6 and miR-33 ASO *n* = 6) and *$P$ < 0.05 comparing miR-33 ASO group with PBS and control ASO group.

K     Representative Western blot analysis of ABCA1, NPC1, SRC1, RIP140, CROT, and NFYC from liver lysates of mice treated with PBS, control ASO, or miR-33 ASO. Tubulin was used as a loading control.

Source data are available online for this figure.

NFYC, SREBP1, ACC, and FASN, was also increased in miR-33 ASO-treated mice (Fig 2F). Moreover, the expression of HMGCR, the rate-limiting enzyme of cholesterol biosynthesis, and the LDLR were also upregulated when we silenced miR-33 in mice (Fig 2G). Interestingly, the expression of ABCA1, a very well-established miR-33 target gene, only increased significantly at the protein level (Fig 2G), suggesting that this gene is likely regulated at the translational level by miR-33. Together, these results demonstrate that long-term anti-miR-33 therapy in mice fed a HFD results in a derepression of numerous miR-33 target genes involved in cholesterol export and

fatty acid oxidation but causes a significant increase in the expression of genes associated with cholesterol and fatty acid synthesis, thus leading to moderate hepatic steatosis.

To investigate whether miR-33 directly regulates genes involved in fatty acid and cholesterol metabolism, we used a combination of bioinformatic approaches (targetscan, pictar, and mirwalk) to identify potential novel targets of miR-33. Interestingly, we found that miR-33 has predicted binding sites in the 3′UTR of *SREBP1* and *HMGCR* (Supplementary Fig S1A). Both predicted binding sites are conserved in mammals (Supplementary Fig S1B). To directly assess

the effect of miR-33 on these predicted target genes, we cloned the 3′UTR of *HMGCR* and *SREBP1* into luciferase reporter plasmids. Surprisingly, we found that miR-33 fails to repress the 3′UTR activity of both genes, suggesting that miR-33 does not regulate their expression directly (Supplementary Fig S1C and D). Taken together, these results suggest that the increased expression of SREBP-responsive genes in mice treated with miR-33 ASO is not due to a direct effect of miR-33 on SREBP1 but rather to its inhibitory action on NFYC, which is a SREBP co-activator.

### Prolonged miR-33 silencing results in significant alterations in enzymes associated with glucose metabolism

Finally, we further explored the impact of long-term miR-33 treatment on hepatic protein expression using a proteomic approach. Liver lysates were obtained from mice treated with miR-33 ASO or Cont ASO and fed a HFD for 20 weeks. We found that proteins involved in glucose metabolism were highly altered in mice treated with miR-33 ASO (Supplementary Table S1). Indeed, a bioinformatic analysis of biological processes shows that the proteins altered in the liver of mice administered with miR-33 ASO were significantly enriched (FDR< 0.001) for the regulation of glucose metabolism as well as other metabolic processes (Supplementary Fig S2). This finding was not unexpected given the role of miR-33 in regulating glucose

metabolism. In addition to proteins involved in glucose homeostasis, we also found a significant enrichment of proteins associated with lipid metabolism, including ApoA1 and HADHB and the post-translational attachment of acetyl groups (Supplementary Fig S2 and Table S1). The most pronounced change, however, was the upregulation of the major urinary protein (MUP) (Fig 3 and Supplementary Table S1). MUP is a secreted protein that is primarily synthesized in the liver. It increases energy expenditure and improves glucose and lipid metabolism in male mice through the paracrine/autocrine regulation of the hepatic gluconeogenic and/or lipogenic pathway (Hui *et al*, 2009; Zhou *et al*, 2009). Taken together, these results demonstrate the complex network of genes that miR-33 regulates. Moreover, they highlight the marked effect that prolonged anti-miR-33 therapy causes in multiple metabolic pathways.

## Discussion

Manipulating the expression of miRNAs *in vivo* has tremendous potential for treating human cardiometabolic diseases, including atherosclerosis, type II diabetes, and metabolic syndrome (Fernandez-Hernando *et al*, 2011, 2013). We and others have previously shown that short-term treatment with miR-33 inhibitors markedly increases plasma HDL-C levels and enhances the

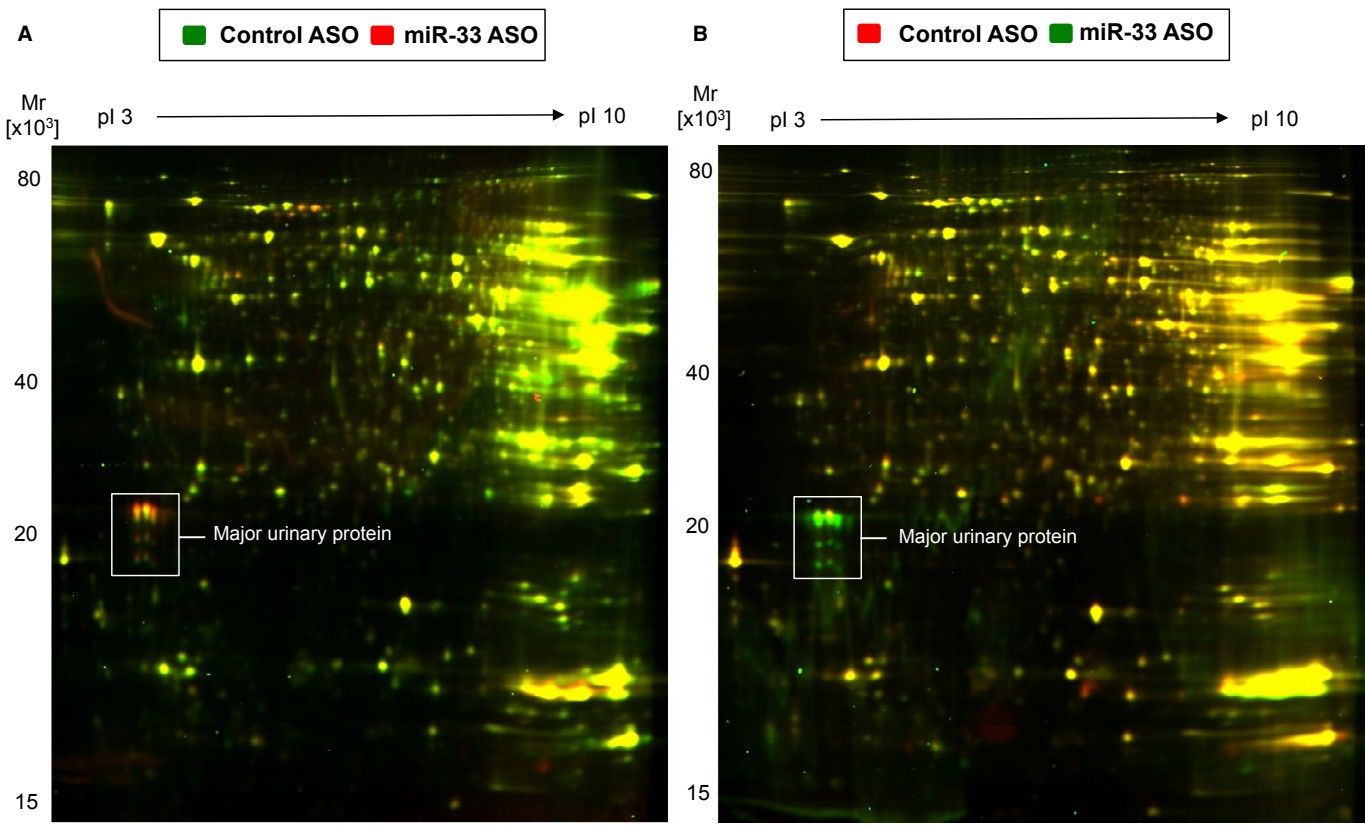

**Figure 3.  Anti-miR-33 therapy causes a profound alteration in the liver proteome.**

A, B   Liver protein extracts from control ASO (green color) and miR33 ASO-treated (red color) mice were quantified using difference gel electrophoresis (DIGE, *n* = 4 per group) (A). Among the differentially expressed spots was major urinary protein (MUP). Note the pronounced upregulation of MUP (white box) in response to miR-33 ASO. Results were reproduced by using a dye-swap (B): control ASO (red color), miR-33 ASO (green color).

regression of atherosclerosis (Marquart *et al*, 2010; Najafi-Shoushtari *et al*, 2010; Rayner *et al*, 2010, 2011a,b; Horie *et al*, 2013). However, the efficacy of anti-miR-33 therapy on the progression of atherosclerosis is controversial (Horie *et al*, 2012; Marquart *et al*, 2013; Rotllan *et al*, 2013). Nevertheless, $miR-33^{-/-}/apoE^{-/-}$-deficient mice developed smaller atherosclerotic plaques than $apoE^{-/-}$ mice (Horie *et al*, 2012). Even though these results are promising for treating cardiovascular diseases, the safety and physiological effect of prolonged miR-33 silencing remains to be elucidated. Here, we demonstrate that long-term treatment with miR-33 ASO in mice fed a CD increases plasma HDL-C levels without any adverse effect. However, when mice were fed a HFD and treated with miR-33 ASO, they developed hypertriglyceridemia and moderate hepatic steatosis, compared to control animals. This latter phenotype was paralleled by a marked increased expression of genes involved in fatty acid and cholesterol synthesis. Importantly, NFYC, a miR-33 target gene and a co-activator of the SREBP genes, was also increased in the livers of mice treated with miR-33 ASO. Together, these data suggest that prolonged anti-miR-33 therapy in mice fed a HFD could be deleterious for treating atherosclerosis and dyslipidemias. Further studies are necessary to understand the complex gene regulatory network controlled by miR-33, as well as the role of miR-33 in regulating metabolism in individual tissues such as the liver, adipose tissue, and brain.

A number of studies have recently identified miR-33 as a potential therapeutic target for treating cardiometabolic disorders including atherosclerosis and metabolic syndrome (Rayner *et al*, 2011b; Horie *et al*, 2012, 2013). These reports demonstrated that miR-33 silencing in mice results in increased circulating HDL-C and bile secretion, thereby enhancing mobilization of sterols accumulated from the peripheral tissue through the reverse cholesterol transport (RCT) pathway (Rayner *et al*, 2011b; Allen *et al*, 2012). Since increased RCT correlates inversely with the incidence of coronary artery disease, several groups studied the efficacy of anti-miR-33 therapy during the progression and regression of atherosclerosis. In the single-regression study published, Moore's group demonstrated that 4-week treatment with 2′F/MOE anti-miR-33 oligonucleotides accelerated the regression of atherosclerosis in $Ldlr^{-/-}$ mice with established atherosclerotic plaques (Rayner *et al*, 2011b). The atherosclerosis progression studies, however, have opposite outcomes. While Baldan's group found that a 12-week anti-miR-33 therapy failed to sustain increased circulating HDL-C and prevent atherogenesis (Marquart *et al*, 2013), we reported that miR-33 ASO successfully reduced the progression of atherosclerosis despite the insignificant alteration of HDL-C levels (Rotllan *et al*, 2013). The discrepancies observed between both atherosclerosis progression studies may be explained by the different oligonucleotide chemistry and slightly different diets used. However, the fact that the genetic ablation of miR-33 protects against the progression of atherosclerosis in $apoE^{-/-}$ mice suggests that long-term anti-miR-33 therapy should be beneficial for treating atherosclerotic vascular disease (Horie *et al*, 2012). The most remarkable difference between the miR-33 antisense therapy and genetic studies is that the ability to increase plasma HDL-C levels was lost in the two progression studies using anti-miR-33 oligos, while $miR-33^{-/-}apoE^{-/-}$ mice still had increased circulating HDL-C. These results suggest that miR-33 ASO delivery may not completely inhibit miR-33 activity in the liver.

In addition to the effect of miR-33 in controlling cholesterol efflux and HDL-C synthesis, miR-33 also regulates the expression of genes involved in fatty acid oxidation and insulin signaling, including *CROT*, *CPT1A*, *HADHB*, *AMPK,* and *IRS2* (Gerin *et al*, 2010; Davalos *et al*, 2011). Therefore, antagonism of miR-33 *in vivo* could potentially represent a novel therapy for treating major risk factors associated with metabolic syndrome including low HDL-C, hypertriglyceridemia, insulin resistance, and non-alcoholic fatty liver disease. Most of the previous studies using miR-33 ASOs were performed over a short period of time and using atheroprone mouse models such as $apoE^{-/-}$ and $Ldlr^{-/-}$ mice. To determine the efficacy of long-term anti-miR-33 therapy in raising plasma HDL-C levels and preventing hepatic steatosis, we administered miR-33 ASO to mice fed a CD (12 weeks) and HFD (20 weeks). Surprisingly, we found that long-term administration of miR-33 ASO results in hypertriglyceridemia and moderate hepatic steatosis. Similar to our results, Horie and colleagues have recently reported that $miR-33^{-/-}$ mice develop obesity, fatty liver, and hypertriglyceridemia. This phenotype was only observed when the mice were fed a HFD but not a CD. Mechanistically, they found that miR-33-deficient mice have increased SREBP1 expression and activation, leading to a transcriptional activation of genes involved in fatty acid synthesis. These results were similar to those reported here using miR-33 ASO. However, we could not identify SREBP1 as a direct target of miR-33 using 3′UTR luciferase experiments. Instead, we found that NFYC, a member of the three NF-Y subunits required for DNA binding and full transcriptional activation of SREBP-responsive genes, was upregulated in the livers of mice administered with miR-33 ASO. In addition to NFYC, we found that SRC1 and RIP140, two transcriptional regulators that control adipogenesis and lipid metabolism, were derepressed in mice treated with miR-33 ASO (Fig 2). Interestingly, SRC3- and RIP140-deficient mice are resistant to obesity and hepatic steatosis, suggesting that the upregulation of these genes observed in anti-miR-33-treated mice might result in lipid accumulation in the liver (Leonardsson *et al*, 2004; Coste *et al*, 2008).

Even though all these studies strongly demonstrate that manipulation of miR-33 levels *in vivo* markedly influences lipid metabolism and atherogenesis, the absence of miR-33b in rodents limits the translational and physiological relevance of these findings. To gain insights into the functional importance of miR-33 in humans, two independent groups assessed the efficacy of inhibiting miR-33 in non-human primates. Treatment of African green monkeys with anti-miR-33 oligonucleotides significantly increased circulating HDL-C (30–40%) in both studies (Rayner *et al*, 2011a; Rottiers *et al*, 2013). However, the effect on plasma TG levels was different. While Moore and Temel's study claimed a marked reduction of plasma VLDL-TGs (Rayner *et al*, 2011a), Näär and colleagues reported that the inhibition of miR-33 in non-human primates does not influence circulating VLDL-TGs (Rottiers *et al*, 2013). In any case, both studies reported that anti-miR-33 therapy does not cause adverse effects, including liver toxicity, as assessed by measuring plasma transaminase levels. This is particularly remarkable in Näär's study where the non-human primates were treated for over 100 days (Rottiers *et al*, 2013). Nevertheless, the adverse effects reported in miR-33-deficient mice and in this study raise awareness that long-term inhibition of miR-33 might cause adverse effects, such us hypertriglyceridemia and hepatic steatosis.

Taken together, our findings demonstrate that long-term pharmacological inhibition of miR-33 leads to dyslipidemia and moderate hepatic steatosis. These findings open new questions about how miR-33 regulates lipid and glucose metabolism at the organismal level. Further studies will be important for elucidating the molecular mechanism and tissue specificity by which miR-33 controls cholesterol, fatty acid, and glucose metabolism.

# Materials and Methods

### Mice

Eight-week-old male C57BL/6 mice were purchased from Jackson Laboratories (Bar Harbor, ME, USA) and kept under constant temperature and humidity in a 12-h controlled dark/light cycle. Mice were randomized into three groups ($n$ = 15 mice): no treatment (PBS), 2′F/MOE anti-miR-33 (TGCAATGCAACTACAATGCAC) oligonucleotide, and 2′F/MOE mismatch control (TCCAATCCAACT TCAATCATC) oligonucleotide (the mismatched bases are underlined). Oligonucleotides were provided from Regulus Therapeutics. Mice received one subcutaneous injection of 5 mg/kg weekly for 12-week chow diet (CD) or 20-week high-fat diet (HFD) consisting of 45% kcal from fat (D12492, Research Diet). Blood samples were collected at 0, 4, 12 and 20 weeks after treatment for lipid analysis and lipoprotein profile measurements. Then mice were sacrificed, and hepatic gene expression and liver histology were analyzed. All animal experiments were approved by the Institutional Animal Care Use Committee of New York University Medical Center and Yale University School of Medicine.

### Lipids, lipoprotein profile, and glucose measurements

Mice were fasted for 12–14 h before blood samples were collected by retro-orbital venous plexus puncture. Then, plasma was separated by centrifugation and stored at −80°C. Total plasma cholesterol and triglycerides were enzymatically measured (Wako Chemicals, USA) according to the manufacturer's instructions. The lipid distributions in plasma lipoprotein fractions were assessed by fast-performance liquid chromatography (FPLC) gel filtration with two superose 6 HR 10/30 columns (Pharmacia).

### RNA isolation and quantitative real-time PCR

Total RNA was isolated using TRIzol reagent (Invitrogen) according to the manufacturer's protocol. For mRNA quantification, cDNA was synthesized using iScript RT Supermix (Bio-Rad), following the manufacturer's protocol. Quantitative real-time PCR (qRT-PCR) analysis was performed in triplicate using iQ SYBR green Supermix (BioRad) on an iCycler Real-Time Detection System (Eppendorf). The mRNA level was normalized to GAPDH or 18S as a housekeeping gene. The human primer sequences used are listed in the Supplementary Table S1. For miRNA quantification, total RNA was reverse transcribed using the miScript II RT kit (Qiagen). Primers specific for human and mouse mmu-miR-33 (Qiagen) were used and values normalized to SNORD68 (Qiagen) as a housekeeping gene. Mmu-miR-33 quantification and quantitative real-time PCR were performed in triplicate using SYBR Green

Master Mix (SA Biosciences) on an iCycler Real-Time Detection System (Eppendorf).

### 3′UTR luciferase reporter assays

cDNA fragments corresponding to the entire 3′UTR of human *SREBP1*, and *HMGCR* were amplified by RT–PCR from total RNA extracted from HepG2 cells with XhoI and NotI linkers. The PCR product was directionally cloned downstream of the *Renilla* luciferase open reading frame of the psiCHECK2™ vector (Promega) that also contains a constitutively expressed firefly luciferase gene, which is used to normalize transfections. All constructs were confirmed by sequencing. COS7 cells were plated into 12-well plates and co-transfected with 1 μg of the indicated 3′UTR luciferase reporter vectors and miR-33 mimics or control mimics (CM) (Life Technologies) utilizing Lipofectamine 2000 (Invitrogen). Luciferase activity was measured using the Dual-Glo Luciferase Assay System (Promega). *Renilla* luciferase activity was normalized to the corresponding firefly luciferase activity and plotted as a percentage of the control (cells co-transfected with the corresponding concentration of control mimic). Experiments were performed in triplicate wells of a 12-well plate and repeated at least three times.

### Hepatic lipid analysis

Liver (50 mg) was homogenized in 500 μl of PBS, and lipids were extracted from 100 μl of the homogenate in the presence of internal standards for each lipid using the Bligh and Dyer method. The different lipid classes (triglycerides, diglycerides, free fatty acids, and cholesterol esters) were quantified from chloroform extracts using shotgun lipidomics based on class separation by MS/MS-specific methods (Allen *et al*, 2012).

### Western blot analysis

Cells were lysed in ice-cold buffer containing 50 mM Tris–HCl, pH 7.5, 125 mM NaCl, 1% NP-40, 5.3 mM NaF, 1.5 mM NaP, 1 mM orthovanadate and 1 mg/ml of protease inhibitor cocktail (Roche), and 0.25 mg/ml AEBSF (Roche). Liver lysates were rotated at 4°C for 1 h before the insoluble material was removed by centrifugation at 12,000 × $g$ for 10 min. After normalizing for equal protein concentration, cell lysates were resuspended in SDS sample buffer before separation by SDS–PAGE. Following overnight transfer of the proteins onto nitrocellulose membranes, the membranes were probed with the following antibodies: ABCA1 (Abcam), NPC1 (Novus), CPT1A, CROT, SRC1, NFYC, RIP140, and Tubulin (1:1,000). Protein bands were visualized using the Odyssey Infrared Imaging System (LI-COR Biotechnology).

### Histology

For our histology analysis, liver samples were fixed in 4% paraformaldehyde (PFA) at 4°C, cryoprotected overnight in 30% sucrose solution, embedded in Tissue-Tek OCT embedding compound, and frozen on dry ice. Next, 8-μm frozen sections were rehydrated, and neutral lipid accumulation was detected by Oil Red-O staining. Sections were rinsed with 60% isopropanol and stained for 20 min with prepared Oil Red O solution (0.5% in isopropanol followed by

dilution to 60% with distilled water and filtered). After two rinses in 60% isopropanol and distilled water, slides were counterstained with hematoxylin for 4 min, rinsed with water, and mounted. Digital images were taken with a Nikon SMZ 1000 microscope connected to a Kodak DC290 digital camera.

## Proteomic analysis

Difference in-gel electrophoresis (DIGE) was performed as described previously (Yin *et al*, JMCC 2013). Briefly, pulverized liver tissue was incubated in 2DE lysis buffer (9.5 M urea, 2% (wt/vol) CHAPS, 0.8% (wt/vol) Pharmalyte pH 3-10, and 1% (wt/vol) DTT plus protease inhibitors) for 0.5 h at RT. After centrifugation at 14,600 *g* for 15 min, the supernatant was collected, proteins were precipitated (2D Clean-up kit; Biorad) and resuspended in DIGE buffer. Protein concentrations were normalized using the Bradford assay. Samples were labeled with fluorescent dyes Cy3 and Cy5 with Cy2 being reserved as the internal standard. Incubation with the dyes was done at a dye/protein ratio of 400 pmol/100 μg for 30 min on ice with the reaction being quenched with 10 mM lysine (L8662, Sigma) for 15 min. Samples were mixed in 2× buffer (8 M urea, 4% (wt/vol) CHAPS, 2% (wt/vol) DTT, 2% (vol/vol) Pharmalyte pH 3-10), and a volume of sample calculated to have a protein content of 50 μg was diluted in rehydration solution (8 M urea, 0.5% (wt/vol) CHAPS, 0.2% (wt/vol) DTT, and 0.2% (vol/vol) Pharmalyte pH 3-10) and loaded onto a IPG strip for isoelectric focusing (18 cm, pH 3-10, GE healthcare) for overnight rehydration. Strips were focused overnight for 64.6 kV hrs using a gradient program at 20°C. Strips were equilibrated and run on a 12% polyacrylamide SDS gel until the blue dye front reached the end of the gel. Fluorescent images of gels were obtained by scanning with an Ettan Dige Image scanner (GE healthcare). Differentially expressed spots showing statistical significance ($P < 0.05$) were identified by using the DeCyder software (Version 7, GE healthcare). Gels were then silver stained (Plus one silver staining kit, GE healthcare) and spots excised for analysis by mass spectrometry. Excised gel spots were subjected to in-gel tryptic digestion with an Investigator ProGest (Genomic Solutions) robotic digestion system with subsequent lyophilization. Freeze-dried samples were resuspended in 20 μl of 0.05% trifluoroacetic acid. Samples were then identified via separation by nano-flow liquid chromatography on a reverse-phase column (Dionex) interfaced to a high-performance linear ion trap mass spectrometer (LTQ XL, Thermo Fisher). Spectra were collected for analysis and searched through mouse protein databases using Mascot and imported into Scaffold (Proteomesoftware). The enrichment analysis of functional annotation terms associated with the differentially expressed proteins was performed using the Database for Annotation, Visualization and Integrated Discovery (DAVID) (Huang *et al*, 2009).

## Statistical analysis

Data are presented as mean ± the standard error of the mean (SEM) (n is noted in the Figure legends). Statistical differences between groups were evaluated using unpaired two-tailed Student's *t*-test, one-way ANOVA and post hoc Tukey's test, or two-way ANOVA with Bonferroni correction for multiple comparisons, as appropriate. Significance was accepted at the level of $P < 0.05$. Data

### The paper explained

#### Problem
Accumulation of cholesterol in the artery wall causes atherosclerosis, the major cause of death in Western societies. HDL-cholesterol (the "good cholesterol") plays a protective role by removing cholesterol from the peripheral tissues to the liver for elimination. Recent studies have demonstrated that specific inhibition of a tiny RNA (miR-33) results in increased circulating HDL levels and prevents against the progression of atherosclerosis. Even though these studies are encouraging, the effect of prolonged miR-33 inhibition has not been studied yet. Therefore, we assessed the long-term efficacy of anti-miR-33 therapy in controlling lipid metabolism.

#### Results
We treated mice fed a chow diet and high-fat diet with miR-33 inhibitors (miR-33 ASO). Unexpectedly, we found that chronic inhibition of miR-33 results in hypertriglyceridemia and moderate hepatic steatosis. These effects appear to be mediated by the upregulation of genes involved in fatty acid synthesis in the liver of mice treated with miR-33 ASOs. Mechanistically, we found that miR-33 inhibition raises NFYC expression. NFYC enhances SREBP transcriptional activation leading to increased expression of fatty acid genes. Proteomic analysis demonstrates that prolonged anti-miR-33 therapy results in marked changes in protein expression. Of note, numerous genes associated with glucose and lipid metabolism were altered in miR-33 ASO-treated mice.

#### Impact
Our findings provide the first evidence that long-term anti-miR-33 therapy results in adverse effects, including hypertriglyceridemia and moderate hepatic steatosis. Further research is required to better understand the complex gene regulatory network controlled by miR-33.

analysis was performed using GraphPad Prism 6.0a software (GraphPad, San Diego, CA, USA).

**Supplementary information** for this article is available online: http://embomolmed.embopress.org

## Acknowledgements
This work has been supported by grants from the National Institutes of Health, R01HL107953, R01HL107953-04S1 and R01HL106063 (to CF-H), 1F31AG043318 (to LG), and R01HL107794 (to AB), and a grant from the Foundation Leducq Transatlantic Network of Excellence in Cardiovascular Research (to CF-H and MM). This work was supported by Heart Research UK and the Department of Health via the National Institute for Health Research (NIHR) Biomedical Research Centre based at Guy's and St Thomas' NHS Foundation Trust and King's College London, in partnership with King's College Hospital NHS Foundation Trust. MM is a Senior Fellow of the British Heart Foundation. RMA was supported by an AHA Predoctoral Fellowship (11PRE7240026).

## Author contributions
LG, AS, AB, MM, and CFH conceived and designed the experiments; LG, AS, CMR, RMA, LG, XY, SL, and AW performed the experiments; LG, AS, CMR, AW, LG, RMA, XY, SL, EAF, YS, AB, MM, and CFH analyzed the data; CE provided control and anti-miR-33 oligonucleotides; LG and CFH wrote the manuscript.

## Conflict of interest
Carlos Fernandez-Hernando has patents on the use of miR-33 inhibitors. C.C. Esau is an employee of Regulus Therapeutics.

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
