## [Review Process File · EMBO Molecular Medicine]

Long-term therapeutic silencing of miR-33 increases circulating triglyceride levels and hepatic lipid accumulation in mice

Leigh Goedeke, Alessandro Salerno, Cristina M. Ramírez, Liang Guo, Ryan M. Allen, Xiaoke Yin, Sarah R. Langley, Christine Esau, Amarylis Wanschel, Edward A. Fisher, Yajaira Suárez,, Angel Baldán, Manuel Mayr and Carlos Fernández-Hernando

Corresponding author: Carlos Fernandez-Hernando, Yale University School of Medicine

Review timeline:

Submission date:	08 March 2014
Editorial Decision:	11 April 2014
Revision received:	12 May 2014
Editorial Decision:	02 June 2014
Revision received:	18 June 2014
Accepted:	26 June 2014

Transaction Report:

Editor: Céline Carret

1st Editorial Decision

11 April 2014

Thank you for the submission of your manuscript to EMBO Molecular Medicine. We have now heard back from the three referees whom we asked to evaluate your manuscript. I am sorry that it has taken so long to get back to you on your manuscript.

While reviewers 1 and 3 delivered their evaluations in a timely manner, we did not receive the other reviewers' input. As the evaluations from the first two reviewers are rather consistent, and a further delay cannot be justified, I have decided to proceed based on these evaluations.

You will see that while both reviewers are generally supportive of your work and underline its potential interest, they also raise a number of specific concerns that prevent us from considering publication at this time. They both request additional experiments to strengthen the main message and provide mechanistic insights but also ask for better explanations and clarifications.

Should you be able to address the raised concerns with additional experiments where appropriate, we would be willing to consider a revised manuscript.

Please note that it is EMBO Molecular Medicine policy to allow a single round of revision in order to avoid the delayed publication of research findings. Consequently, acceptance or rejection of the manuscript will depend on the completeness of your responses included in the next final version of the manuscript.

I look forward to seeing a revised form of your manuscript as soon as possible.

***** Reviewer's comments *****

Referee #1 (Comments on Novelty/Model System):

The authors need to show the data points for all 3 experimental groups, but the findings could be very important.

Referee #1 (Remarks):

This manuscript by Goedeke et al. describes the effect of longterm inhibition of miR-33 during a high fat diet in mice. While the authors confirm the previously published effects on HDL, they also show that longterm inhibition of miR-33 also leads to adverse effects like an increase in triglycerides and accumulation of lipids in the liver.

These findings are very interesting in the fact that they underscore the importance of longterm studies to determine the safety of anti-miR-based inhibition of microRNAs. However, because of the relevance of these findings some experimental data should be supplemented.

Specific comments:

- Is there any chance these data might be due to the chemistry use for the anti-miR-33? Although the control consists of comparable chemical modifications it is very well known that the sequence and order of chemical modifications do influence the biological effects of the compounds.
- The authors should explain what the control compound is in more detail.
- In the preclinical anti-miR studies people are currently using amounts of anti-miR that are in far excess of the clinical dose. Could it be possible that lowering of the dose would remove the detrimental effects? Could it be chemistry related?
- Why did the authors put animals on a normal chow diet for 12 weeks and used a 20 weeks diet scheme for the animals on the high fat diet? For consistency these time point should be comparable.
- Can the authors comment on the influence of stress in these studies? Why would the adverse events only occur when animals are putten on a high fat diet?
- In all 3 figures the data points should be shown for PBS, anti-miR-33 and control and not for anti-miR-33 and control alone. For some data points there seems to be an intermediate effect with the control compound.

Some minor points:

- In the intro the Osborne 2000 ref seems out of place
- In the intro it says: 'Recently we and others have previously identified.'. Either Recently or previously needs to go.
- In the first part of the results section it reads anti-mir-33 i.o. anti-miR-33
- In the second paragraph of the results section it reads pricrossirius red

Referee #3 (Remarks):

In this manuscript, Goedeke et al. studied the effects of long-term treatment with anti-miR-33 oligonucleotides in mice. They found that in addition to HDL which has been determined in previous studies, circulating triglyceride levels was increased as well by silencing miR-33 in the mice fed a high-fat diet but not chow diet. Moreover, lipid was accumulated in the liver after long-term treatment in these mice. The genes involved in fatty acid synthesis and glucose metabolism were increased in the liver by prolonged miR-33 silencing. They concluded that because of the

potential deleterious effects, the application of anti-miR-33 treatment of atherosclerosis should be assessed carefully. Because miRNAs regulate the expression of more than one target and are involved in very complicated networks in most cases, this reviewer agrees that the miRNA-related therapeutic strategy must be assessed seriously. However, some issues should be further addressed in this manuscript. Detailed comments are provided below.

Major comments:

1. The authors used 5 mg/kg as the dose of 2'F/MOE anti-miR-33 oligonucleotide weekly for the treatment in mice in this manuscript. How did the authors decide the dosage? In their previous paper (J Clin Invest. 2011;121:2921-31), they used the dosage of 10 mg/kg. Why didn't they use the same treatment in this manuscript as before? That would be more comparable between short-term and long-term treatment.
2. In Figure 1C and I, no effects were observed at 4-week time point. This is not consistent with their previous publication (J Clin Invest. 2011;121:2921-31, Science. 2010;328:1570-3).
3. In Figure 1G, 2A and 2D, the PBS groups which are indicated in the main text are missing.
4. In Figure 1J, circulating TG level was increased not only after long-term treatment (12 and 20 weeks) but also after short-term treatment (4 weeks). Therefore, it's not precise to emphasize that long-term, but not short-term treatment with anti-miR-33 oligonucleotide brought adverse effect.
5. The deleterious effects were only observed in the mice fed an HFD, but not with CD feeding. To this reviewer, HFD feeding during treatment period does not make sense because diet control is necessary during treatment period of patients with atherosclerosis.
6. ABCA1 expression was only affected at protein level, but not mRNA level, by silencing miR-33 (Figure 2G). The authors suggest that ABCA1 is likely be regulated at translational level by miR-33. This is not consistent with the previous studies in which they found ABCA1 mRNA was regulated by miR-33 (J Clin Invest. 2011 Jul;121(7):2921-31).
7. The authors concluded that the increased expression of SREBP-responsive genes in mice treated with miR-33 ASO is not due to a direct effect of miR-33 on SREBP1 but rather to its inhibitory action on NFYC, which is a SREBP co-activator, which is overstated and is not supported by sufficient experimental evidence. For example, does silencing NFYC interrupt the effects of miR-33 ASO on the expression of SREBP-responsive genes?

Minor comment:

1. The authors referred to the wrong figure about mRNA expression of HMGCR, LDLR and ABCA1. It should be Fig 2G, but the authors referred to Fig 1G.

1st Revision - authors' response

12 May 2014

RESPONSE TO THE REVIEWERS

Reviewer #1

We would like to thank the reviewer for his/her comments, which truly helped us to improve our study. In the revised version of the manuscript we have addressed all the reviewer's concerns.

Specific comments

A. Comments on Novelty/Model System:

1) "The authors need to show the data points for all 3 experimental groups, but the findings could be very important".

As suggested by the reviewer, we have included the data for all experimental groups. As seen in the new **Figure 1** and **2**, Cont ASO treated mice have similar plasma lipid levels and hepatic lipid contents than mice injected with PBS (vehicle). Moreover, the hepatic mRNA and protein expression analysis of genes associated with lipid and glucose metabolism was similar in both control groups of mice.

Since the effects on lipoprotein metabolism, hepatic lipid composition and gene expression were similar between both groups of mice; we only performed the proteomic analysis in mice treated with miR-33 ASO and Cont ASO.

2) *Remarks:*

We thank the reviewers her/his positive comments (“***These findings are very interesting in the fact that they underscore the importance of long-term studies to determine the safety of antimiR-based inhibition of microRNAs***”).

Specific comments:

1) “***Is there any chance these data might be due to the chemistry use for the antimiR-33? Although the control consists of comparable chemical modifications it is very well known that the sequence and order of chemical modifications do influence the biological effects of the compounds***”.

We are aware that some reports have shown some non-specific effects of antisense oligonucleotides but we think that it would be very unlikely this occurred in our study. The chemical modification used in the Cont ASO and miR-33 ASO was the same and the sequence only varies in 4 nucleotides (see material and methods in the new version of the manuscript). Moreover, we had previously used both antisense oligonucleotides and did not find any signs of hepatic toxicity or adverse effects (Rayner KJ, Nature 2011). Lastly, a recent publication by Horie et al (Nature Communications, 2014) showed that miR-33 null mice fed a high-fat diet (HFD) developed hepatic steatosis, similar to our findings following therapeutic silencing of the miRNA, thus suggesting that the effects of long-term miR-33 silencing reported here are specific.

2) “***The authors should explain what the control compound is in more detail***”.

As suggested by the reviewer, we have included a detailed description of the control compound used in this study (“*Regulus therapeutics provided us the 2’F/MOE-modified, phosphorothioate-backbone-modified antisense miR-33 (TGCAATGCAACTACAATGCAC) and mismatch control (TCCAATCCAATTCATC) antimiRNA (the mismatched bases are underlined)*”).

3) “***In the preclinical anti-miR studies people are currently using amounts of anti-miR that are in far excess of the clinical dose. Could it be possible that lowering of the dose would remove the detrimental effects? Could it be chemistry related?***”

This is a good question. Most of the preclinical studies have been performed at shorter time points using higher doses (10 mg/Kg). It would be interesting to perform a number of studies using different doses and time points. However, the fact that miR-33 deficient mice showed a similar phenotype that mice treated with miR-33 ASOS suggest that the effect is target-related, rather than specific to the anti-miR chemistry. Further studies will be important to assess whether optimizing the dosing regimen of miR-33 inhibitors *in vivo* can mitigate these findings.

4) “***Why did the authors put animals on a normal chow diet for 12 weeks and used a 20 weeks diet scheme for the animals on the high fat diet? For consistency these time point should be comparable***”.

We apologize to the reviewer for this misunderstanding. When we treated mice on chow diet, we did not observe differences in plasma TG levels after 1, 2 and 3 months (**Figure. 1E**). However, mice fed a HFD showed a significant increase in circulating TG after one month of treatment (**Figure. 1J**). Therefore, we decided to extend the study of the effect of miR-33 inhibition in mice fed a HFD for an extra two-month period (12-week time point in **Figure. 1J**). Additionally, we were initially interested in determining whether miR-33 antagonism might prevent hepatic steatosis, since previous results from our laboratory (Davalos A, PNAS 2011) and from Gerin *et al* (JBC, 2010) showed that miR-33 controls the expression of genes involved in hepatic fatty acid b-oxidation. Thus, we extended the HFD feeding for a total of 20 weeks instead of 12, to better assess hepatic

steatosis. Surprisingly, we found that prolonged miR-33 inhibition results in hepatic lipid accumulation. A extended discussion about these results has been incorporated in the revised version of the manuscript (page 11)

5) “Can the authors comment on the influence of stress in these studies? Why would the adverse events only occur when animals are putten on a high fat diet?”

Based on previous reports from our lab (PNAS, 2011) and from Gerin *et al* (JBC, 2010) that showed miR-33 controlling the expression of genes involved in fatty acid b-oxidation, one would expect that silencing of miR-33 could ameliorate both liver TG accumulation and plasma TAG levels due to accelerated hepatic lipid catabolism. Hence, the finding that anti-miR-33 oligos resulted in increased hepatosteatosis and hypertriglyceridemia were surprising. Interestingly, this phenotype occurred only in mice fed a HFD, but not on chow-fed animals. While the reasons for this are obscure, we hypothesize that additional, yet-unknown targets of miR-33 that influence the lipogenic pathway and overcome the effects on b-oxidation become preferentially de-repressed in antimir-33 mice only under conditions of excess lipid influx (i.e. following a HFD regimen). Interestingly, Horie *et al* recently reported a similar phenotype in miR-33 KO mice, which developed hepatic steatosis when fed a HFD, but not on chow (Nature Communications, 2014). These authors showed that the levels of Srebp1c become de-repressed in the former mice, but not in the latter, and proposed that Srebp1 is indeed a direct target of miR-33 in vivo under specific dietary conditions. Our own data (**Figure. E1**), and those from Allen *et al* (EMBO Mol Med 2012), however, failed to show a direct targeting of Srebp1 by miR-33. Additional experiments will be necessary to decipher the exact molecular causes behind the diet-specific changes in hepatic lipid metabolism following therapeutic miR-33 silencing.

6) “In all 3 figures the data points should be shown for PBS, antimir-33 and control and not for antimir-33 and control alone. For some data points there seems to be an intermediate effect with the control compound”.

As suggested by the reviewer, we have included the data for all experimental groups in **Figure 1** and **2**. The results demonstrate that Cont ASO treated mice have similar plasma lipid levels and hepatic lipid contents than mice injected with PBS (vehicle). Moreover, the expression of genes associated with lipid and glucose metabolism was similar in both groups of mice.

Since the effects on lipoprotein metabolism, hepatic lipid composition and gene expression were similar between PBS and Cont ASO treated mice, we only performed the proteomic analysis in mice treated with miR-33 ASO and Cont ASO.

Some minor points:

1) “In the intro the Osborne 2000 ref seems out of place”.

We apologize for this oversight that has been corrected in the revised manuscript.

2) “In the intro it says: ‘Recently we and others have previously identified..’. Either Recently or previously needs to go”.

Thank you for catching this. We have corrected it in the revised manuscript.

3) “In the first part of the results section it reads anti-mir-33 i.o. anti-miR-33”

The text has been modified according to the reviewer’s suggestion.

4) “In the second paragraph of the results section it reads pricosirius red”

We apologize for this oversight that has been corrected in the revised manuscript.

Reviewer #3

We would like to thank the reviewer for his/her comments about our study that has helped us to improve our study. In the revised version of the manuscript we have addressed all the reviewer's concerns.

*Specific comments**Major comments:*

1) ***“The authors used 5 mg/kg as the dose of 2'F/MOE anti-miR-33 oligonucleotide weekly for the treatment in mice in this manuscript. How did the authors decide the dosage? In their previous paper (J Clin Invest. 2011;121:2921-31), they used the dosage of 10 mg/kg. Why didn't they use the same treatment in this manuscript as before? That would be more comparable between short-term and long-term treatment”.***

The present study and our previous work are different. Here, we used C57BL6 mice and in the JCI study we analysed the efficacy of miR-33 antagonism in low-density lipoprotein receptor (LDLR) deficient mice. Moreover, we know now that 5 mg/kg is enough to inhibit miR-33 expression in the liver. In the JCI study, we performed a short-term treatment (1 month) to assess the efficacy of anti-miR-33 therapy during atherosclerosis regression. We used a higher dose to increase the efficacy of miR-33 inhibition in the arterial wall.

2) ***“In Figure 1C and I, no effects were observed at 4-week time point. This is not consistent with their previous publication (J Clin Invest. 2011;121:2921-31, Science. 2010;328:1570-3)”.***

In this study, we found that mice treated with miR-33 ASO have significantly more plasma HDL-cholesterol after 12 weeks of treatment compared with Cont ASO or PBS treated mice. Even though we found a positive trend at 4 weeks, the changes in circulating HDL-C between the different groups of mice did not reach statistical significance. This could be explained by the different strain of mice used in the JCI study (*ldlr*^{-/-} mice) or treatment in the Science study (lentivirus vs 2'F/MOE antisense oligonucleotides). Moreover, the high fat diet is an additional variable that may explain the difference.

3) ***“In Figure 1G, 2A and 2D, the PBS groups which are indicated in the main text are missing”.***

We apologize to the reviewer for this oversight. The effect of PBS on hepatic miR-33 levels, lipoprotein metabolism, hepatic lipid content and mRNA and protein expression in the liver have been included in the revised version of the manuscript (new **Figure 1** and **2**).

4) ***“In Figure 1J, circulating TG level was increased not only after long-term treatment (12 and 20 weeks) but also after short-term treatment (4 weeks). Therefore, it's not precise to emphasize that long-term, but not short-term treatment with anti-miR-33 oligonucleotide brought adverse effect”.***

We agree with the reviewer. The text has been modified according to the reviewer's suggestion. Importantly, plasma TG levels were increased in the anti-miR-33 group even at early time points. These results are consistent with a recent report from Allen *et al* (Circ Res, 2014) that showed elevated hepatic VLDL secretion in mice dosed with anti-miR oligos. However, our data show changes in circulating TG only in HFD-fed mice, but not on chow-fed animals.

5) ***“The deleterious effects were only observed in the mice fed an HFD, but not with CD feeding. To this reviewer, HFD feeding during treatment period does not make sense because diet control is necessary during treatment period of patients with atherosclerosis”.***

We agree with the reviewer but we think it would important to emphasize that the anti-miR-33 therapy cause adverse effects only in mice fed a HFD. Additionally, several epidemiological studies

have shown a less-than-ideal degree of dietary compliance in patients diagnosed with CVD, especially those engaged in primary prevention interventions. The development of anti-miR-33 based therapies for the management of hyperlipidemias and the treatment of CVD has gained traction in the last 4 years, not only among academia but also in the pharmaceutical industry. We sincerely believe that the data presented herein are of interest from the basic science perspective and have clear clinical implications, accentuating the notion that caution should be taken with the development of anti-miR-33-based therapies.

6) ***“ABCA1 expression was only affected at protein level, but not mRNA level, by silencing miR-33 (Figure 2G). The authors suggest that ABCA1 is likely be regulated at translational level by miR-33. This is not consistent with the previous studies in which they found ABCA1 mRNA was regulated by miR-33 (J Clin Invest. 2011 Jul;121(7):2921-31)”.***

We always observed a modest effect on ABCA1 mRNA levels in WT mice treated with miR-33 ASO compared to Cont miR treated mice. As mentioned above, we have previously assessed the efficacy of silencing miR-33 in LDLR deficient mice. It is possible that the effect of miR-33 ASO on ABCA1 mRNA expression might be influenced by the differences in mouse genetic strains. Another explanation is that LDLR deficiency might increase ABCA1 mRNA expression through another different mechanism, thus allowing for a cleaner de-repression effect of the anti-miR-33 oligos. Nevertheless, the elevated HDL-cholesterol levels noted in the anti-miR group validate the increase in ABCA1 expression in those mice, compared to the control groups.

7) ***“The authors concluded that the increased expression of SREBP-responsive genes in mice treated with miR-33 ASO is not due to a direct effect of miR-33 on SREBP1 but rather to its inhibitory action on NFYC, which is a SREBP co-activator, which is overstated and is not supported by sufficient experimental evidence. For example, does silencing NFYC interrupt the effects of miR-33 ASO on the expression of SREBP-responsive genes?”***

We agree with the reviewer comment about overstating our conclusions. Thus, we modified this part of the manuscript accordingly. Moreover, we have performed the experiment suggested by the reviewer in a mouse hepatoma cell line (Hepa). As shown in **Figure to the Reviewer**, inhibition of NFYC did not inhibited the increased expression of fatty acid synthase (FASN) in cells transfected with anti-miR-33 oligonucleotides, suggesting that additional mechanisms might be involved in increasing SREBP-target genes expression (e.g. FASN) in hepatic cells transfected with miR-33 inhibitors.

Minor comment:

1) ***“The authors referred to the wrong figure about mRNA expression of HMGCR, LDLR and ABCA1. It should be Fig 2G, but the authors referred to Fig 1G. 　”;***

We apologize for this oversight that has been corrected in the revised manuscript.

Thank you for the submission of your revised manuscript to EMBO Molecular Medicine. We have now received the enclosed reports from the referees that were asked to re-assess it. As you will see the reviewers are now supportive and I am pleased to inform you that we will be able to accept your manuscript pending editorial final amendments.

Please submit your revised manuscript within two weeks. I look forward to seeing a revised form of your manuscript as soon as possible.

***** Reviewer's comments *****

Referee #1 (Remarks):

This is a greatly enhanced version of the manuscript and I think the findings are insightful for readers interested in miRNA therapeutics